# Effect of Carbon Content on Mechanical Properties of Boron Carbide Ceramics Composites Prepared by Reaction Sintering

**DOI:** 10.3390/ma15176028

**Published:** 2022-09-01

**Authors:** Wenhao Sha, Yingying Liu, Yabin Zhou, Yihua Huang, Zhengren Huang

**Affiliations:** 1State Key Laboratory of High-Performance Ceramics and Superfine Microstructure, Shanghai Institute of Ceramics, Chinese Academy of Sciences, No. 588, HeShuo Road, Jiading District, Shanghai 201800, China; 2University of Chinese Academy of Sciences, Beijing 100049, China; 3College of Materials Science and Engineering, Nanjing Tech University, Nanjing 211816, China

**Keywords:** composites, elastic modulus, flexural strength, reaction sintering, vickers hardness

## Abstract

In this study, different reaction-bonded boron carbide (RBBC) composites with a free carbon addition from 0 to 15 wt% were prepared, and the effect of the carbon content on the mechanical properties was discussed. With the free carbon addition increase from 0 to 15 wt%, the residual silicon content in the RBBC composite decreased first and then increased. Meanwhile, the strength of the RBBC composite improved first and then worsened. In the RBBC composite without free carbon, the B_4_C grains are obviously dissolved, the grains become facet-shape, and the grain boundary becomes straight. The microstructure of the composite was tested by SEM, and the phase composition of the composite was tested by XRD. The RBBC composite with the addition of 10 wt% free carbon has the highest flexural strength (444 MPa) and elastic modulus (329 GPa). In the composite with a 10 wt% carbon addition, the phase distribution is uniform and the structure is compact.

## 1. Introduction

Boron carbide (B_4_C) ceramics are a significant structural material. They have the advantages of a high hardness, low density, high melting point, excellent corrosion and wear resistance, good chemical stability and neutron absorption performance. They are widely used as bulletproof material, radiation proof material, wear-resistant and self-lubricating material, special acid and alkali-resistant material cutting and grinding tools, and atomic reactor control and shielding materials. However, the sintering and densification of B_4_C ceramics are difficult. Hot pressing can be used for preparing dense B_4_C [1], but only samples with simple shapes can be prepared. There have been many studies on B_4_C prepared by spark plasma sintering in recent years [2,3,4]. However, only small-size samples can be prepared by spark plasma sintering. Pressureless sintering can also prepare dense B_4_C but requires a high temperature (2000 °C) [5,6,7]. Then, these common sintering methods often require a higher cost or higher sintering temperature. In recent decades, reaction sintering has been widely used in the preparation of B_4_C ceramic composites [8]. This new method has two outstanding advantages in the preparation of RBBC composites. First, the sintering temperature of reaction sintering (1450–1650 °C) is far lower than that of pressureless sintering (2000 °C) or hot press sintering. The use of reactive sintering can effectively reduce the sintering temperature. Second, reaction sintering is a net size sintering technology, and the sample size is almost unchanged before and after sintering (size change < 1%) [9].

In recent decades, reaction sintering has attracted more and more attention in the preparation of dense boron carbide ceramics. Adding different contents of free carbon into the material system will obviously affect the structure and properties of reaction-sintered B_4_C ceramics. At present, there are few studies on this aspect, or some aspects have not been mentioned. Zhang et al. [10] reported the preparation of RBBC composites, but did not systematically study the effect of the free carbon content. Gao et al. [11] explored the influence of free carbon content upon mechanical properties, but did not consider the dissolution of the B_4_C main phase in silicon.

In this study, the effect of the amount of free carbon in the material on the mechanical properties and the microstructure evolution during siliconization were investigated. The change of the residual silicon content in RBBC ceramics is also mentioned in this study. In addition, this study provides a new recipe to better improve the properties of boron carbide-based ceramic composites.

## 2. Materials and Methods

### 2.1. Materials

The raw powders were commercially available B_4_C (D_50_ = 20 μm, Mudanjiang diamond boron carbide Co., Mudanjiang, China), amorphous carbon black (D_50_ = 1 μm, Fusman, Beijing, China) powders. The binder used is phenolic resin (Aladdin, Shanghai, China). N-butanol and paraffin wax (Aladdin, Shanghai, China) were also used in this study. Figure 1 shows the particle morphology and the particle size distribution of the raw B_4_C powders. B_4_C and carbon powder, along with a certain amount of binders, N-butanol, paraffin wax were mixed in various proportions. The binder is phenolic resin, and the addition amount is 10% of the sum of the mass of boron carbide powder and silicon carbide powder. N-butanol can eliminate bubbles in slurry. Paraffin is conducive to the complete demoulding process of the sample. Table 1 lists the name of samples and their composition. The mixtures were ball-milled in an ethanol solvent with SiC balls for 24 h. The milling speed was 60 rpm. Then, the slurries were dried at 70 °C for 24 h. The powder mixtures were uniaxially pressed at 200 MPa. The compacted specimens (40 mm × 8 mm × 6 mm) were placed into a vacuum furnace and heated up to 900 °C for half an hour to burn off the binder. After that, the green bodies were infiltrated by silicon from lumps (1–5 mm) placed on the surface of the green compacts. The mass of added silicon lumps is about 1.2 times that of green. The sintering temperature, holding time, and heating rate are 1600 °C, 30 min, and 5 °C /min, respectively. The samples were sintered under vacuum (50 Pa).

### 2.2. Characterization

The Archimedes principle (GB-T1966-1996) was used for measuring the open porosity and density of sintered bodies. The pore size distribution and the open porosity of the green body was measured by Automatic mercury porosimeter (Poremaster60, Anton paar, Ashland, VA, USA). The particle size distribution of the powder was measured by a laser diffraction particle size analyzer (Mastersizer 3000, Malvern, UK). The mechanical properties of sintered samples were tested by a universal material testing machine (Instron-5566, Instron Co., Norwood, MA, USA). The Vickers hardness of samples was tested under a load of 10 N for 10 s and tested for at least 5 independent points. Bars of 3 mm × 4 mm × 36 mm were prepared for the flexural strength test. A three-point bending method (bend span = 20 mm, load speed = 0.5 mm/min) was used for measuring the flexural strength. An indentation method (the applied load = 10 N) was used for measuring the fracture toughness. The microstructure and element distribution of the composites were examined with a scanning electron microscope (SEM, Magellan 400, FEI, Hillsboro, OR, USA) and transmission electron microscope (TEM, JEM-2100F, JEOL, Tokyo, Japan). The surface conditions of the SEM specifications are polished. The polished samples were prepared according to standard metallographic procedures including the final polishing stage with a 0.5 μm diamond paste. The phase was analyzed via X-ray diffraction (XRD, D/Max2250V, Rigaku, Tokyo, Japan) and Energy Dispersive Spectroscopy (EDS). Jade was used to analyze the XRD results. The XRD data was obtained from the polished exterior surfaces of the samples. The residual silicon content of composites was tested by chemical method. The residual silicon in the RBBC composites was extracted by nitric acid and hydrofluoric acid. Then, the content of silicon was determined by potassium fluosilicate volumetric method.

## 3. Results and Discussion

### 3.1. Phase Compositions of RBBC Ceramic Composites

The result of the XRD test is shown in Figure 2. The diffraction peak of B_4_C can be observed in Figure 2a. However, there is no diffraction peak from C in Figure 2a due to the added C being amorphous. 

Figure 2b shows that the main phase composition includes B_13_C_2_, B_12_(C,Si,B)_3_, SiC, and Si. The reaction between B_4_C and Si leads to the formation of B_12_(C,Si,B)_3_ and SiC [12]. The stoichiometric ratio of boron carbide is unstable. From Figure 2a,b, we can see that the diffraction peaks of B_4_C and B_13_C_2_ are almost the same. The sintered samples should still have a large amount of B_4_C phase. Comparing the test results of XRD with the standard XRD card, it can be observed that the generated SiC is 6H-SiC.

In addition, the relative intensity of the main diffraction peak of Si in the four RBBC composites (2θ = 28.6° and 47.5°) shows a trend of first decreasing and then increasing. The main diffraction peak of Si is the lowest when the carbon content is 10 wt%, which indicates that the residual silicon content in the C10 sample is the lowest. The test results of the residual silicon content of the composites also confirmed this conclusion.

Table 2 shows the content of residual silicon in different sintered samples. After sintering, there is still some unreacted silicon in the sample. When the added carbon content changes in the range of 0–10 wt%, the residual silicon in the sintered sample gradually decreases with the increase of the carbon content. In the sintering process, the generated silicon carbide and molten silicon are filled in the pores of the green body [8,13]. Table 3 shows that the open porosity of the four green billets is similar. With the increase of free carbon, the content of silicon carbide increases, and so the content of residual silicon decreases. When the added carbon content is 15 wt%, the raw boron carbide particles dissolved and reacted [14]. The phase of boron carbide in the composite was reduced. Then, more liquid Si entered the sample, and the content of residual silicon increased. The change in the residual silicon and silicon carbide content can also be observed in the SEM (Figure 3). The residual silicon content in the C10 sample is low, which is one of the reasons for its high strength.

### 3.2. Microstructures of RBBC Ceramic Composites

The SEM photographs of four samples are shown in Figure 3. It can be observed in Figure 3a that there are three phases in the RBBC composites. The black, gray, and white areas were B_4_C phase, SiC phase, and free Si, respectively. In Figure 3, the areas of phase became smaller and the distributions of phase become more uniform for carbon content ranging from 0 to 10 wt%. Figure 3a shows the phase distribution of the C0 specimen without free carbon. The phase distribution is nonuniform. The nonuniform distribution may be due to the difficulty of B_4_C sintering, which may lead to a lower strength [15]. The abnormal phase size distribution becomes the flaw size and hence reduces the strength. Previous studies also show that when the microstructure of the composites is more uniform, the strength is higher [16]. 

The formation of more bright, white silicon phases can also be observed in Figure 3d. This shows that the residual silicon content in the sample increases. The increase of the residual silicon content will lead to the decrease of the material strength [17].

From Figure 3, we can see that some abnormally large SiC grains are generated in the sample. In Figure 3a in particular, there are many such large grains. In order to find out the reason for the formation of large SiC grains, the samples were tested by TEM. Figure 4 shows the results of the TEM and energy spectrum tests. The sample used for the TEM test uses a focused ion beam. The area where the sample is cut for the TEM test is indicated in Figure 4a. The sample morphology used for the TEM test is shown in Figure 4b. It can be seen from Figure 4c that the grain boundaries of large grains are obvious. Three regions were selected for energy spectrum point scanning. Spectrum 1 is inside the grain; Spectrum 2 is at the grain boundary; and Spectrum 3 is outside the grain. The results of the energy spectrum point scan are shown in Figure 4d. Large grains are surrounded by Si. B_12_(CxSiyBz) was formed on the surface of large grains, and a small amount of boron was observed inside. During the process of reaction sintering, boron carbide will react with silicon according to Si(l) + 3B_4_C(S) = SiC(S) + B_12_(B,C,Si)_3_(S) [18,19]. B_12_(CxSiyBz) is the product of the reaction between B_4_C and Si. Therefore, we can speculate that the large-size grains are related to the dissolution of B_4_C. B element was observed in the interior of large grains, but because the TEM test area was small, we did not obtain the element composition in the interior of large grains. We put forward a guess about the formation of large grains. Large size grains are formed by the reaction of B_4_C particles with Si in the process of reaction sintering. The reaction first occurred on the surface of B_4_C particles. The surface of the grain is reaction-formed silicon carbide, and there may be unreacted boron carbide phase inside the grain. In the C0 sample, only a small amount of C is provided by phenolic resin, so more boron carbide particles react with silicon and generate more large grains. Additional free C was added to the other three samples, and silicon reacted with C first. Therefore, less boron carbide is reacted and fewer large particles are generated.

Figure 5 shows the EDS data of the polished surface of four samples. As mentioned earlier, the black, gray, and white areas were B_4_C phase, SiC phase, and free Si, respectively. EDS test data also confirmed this result. It is observed that there is a small amount of aluminum in the sample, which may be the impurity introduced in the sintering process.

Figure 6 shows the microstructures of four samples. The black particles were B_4_C grains. The shape of B_4_C grains in the C10, C15 composite (Figure 6c,d) is irregular, and the boundaries are curved. It is consistent with the original boron carbide grains (Figure 1a). However, when the addition of free carbon is reduced to 5 wt%, the shape of the B_4_C grains is changed. It is not irregular, but faceted. In addition, the boundaries of the B_4_C grains become straight. When the addition of free carbon is reduced to 0 wt%, almost all of the B_4_C grains in the C0 sample (Figure 6a) become facet-shaped, and the boundaries become straight. The change in the B_4_C grain may be due to dissolution. During sintering, B_4_C grains dissolve and react in molten silicon [20]. When there is no free carbon in the raw material, the dissolution of B_4_C particles is most serious during the Si infiltration. The serious dissolution of B_4_C grains in the C0 composite results in the mechanical properties of the RBBC composite becoming poor. 

Figure 7 shows the fracture surfaces’ morphology and phase distribution of the composite with different carbon contents. The secondary electron images show the fracture surface morphology, and the backscattered electron images show the phase distribution. The phase distribution of the fracture surface is similar to that of the surface (Figure 3). The black, gray, and white areas were B_4_C phase, SiC phase, and free Si, respectively. The fracture mode of the four composites is transgranular fractures. River-like pattern fracture surfaces with a large-size grain area can be observed in Figure 7a,d. These fracture surfaces show the typical fracture morphologies of brittle materials. This also confirms that the formation of a large-size grain region leads to the reduction of the RBBC composites’ strength. In Figure 7c, one can observe that small-size silicon carbide can surround large-size B_4_C particles. This further shows that the phase distribution in the C10 RBBC composites is uniform.

### 3.3. Density and Mechanical Properties of RBBC Ceramic Composites

#### 3.3.1. Pore Size Distribution and Porosity of Green Bodies

Figure 8 shows the pore size distribution of green bodies with different carbon contents. All green bodies only have one peak. When the carbon content increases, the pore size of the green body gradually decreases; this is due to the smaller amorphous carbon particles filling the pores between the larger B_4_C particles and the pores of the green bodies becoming smaller. The peak of the green body without free carbon (C0) is sharper. This shows that the pore size distribution of the green body without free carbon is more concentrated. This result is assumed to show that the green body only exists for B_4_C particles with the same pore size distribution. In addition, the B_4_C particle size range is relatively narrow, as presented in Figure 1b. Thus, the pore size would also be narrow. 

Table 3 shows the open porosity of four green bodies. The porosity of four green bodies is almost the same and is close to 30%. The green body is porous. Therefore, liquid silicon can easily penetrate into the green body during sintering.

#### 3.3.2. Density and Porosity of RBBC Composites

Figure 9 shows the open porosity and volume density of four samples. The open porosity of four samples is low and less than 1%. There are two reasons for the low porosity of samples. First, during sintering, molten silicon reacted with carbon or boron carbide to form SiC. SiC filled some of the pores [21]. Second, molten silicon entered the remaining pores. The low porosity of the sample indicates that enough liquid silicon entered the green body.

#### 3.3.3. Mechanical Properties of RBBC Composites

The flexural strength and elastic modulus of four samples are shown in Figure 10. Flexural strength and elastic modulus are significant mechanical properties of B_4_C. With the increase of the carbon content, the flexural strength and elastic modulus of the composites have a similar change trend. When the free carbon content increases in the range of 0~10 wt%, the density of sintered samples increases slightly and the flexural strength increases. This is because the defect size in the C10 sample is smaller. The defect size should be the SiC-Si-B_4_C interfaces, based on the exaggerated SiC grain growth and the thermal expansion mismatch between the phases upon cooling. A large-size grain was formed in the C0 and C5 samples, but the phase distribution was uniform in the C10 samples. Therefore, the defects in the C10 sample are smaller. When the free carbon content increases to 15%, the density of the composites changes little. But the phase distribution becomes uneven. This is the reason for the decrease in strength in the C15 sample. The elastic modulus of the material is related to the type of chemical bond. The elastic modulus of B_4_C is much larger than that of Si. The C0 and C15 samples have a high residual silicon content, so the elastic modulus is lower.

The Vickers hardness and fracture toughness of four samples is shown in Figure 11. With the change in the carbon content, the hardness of the material does not change significantly. The hardness of the four samples is stable in the range of 20–22 GPa. The fracture toughness of different RBBC composites is also similar. This may be because the grain sizes of the four samples are similar. The fracture mode of the four composites is transgranular fractures. This may lead to a poor toughness of the sample.

In summary, when the carbon content is 10 wt%, the phase distribution is more uniform and the density is higher. The RBBC composite with the addition of 10 wt% free carbon has the highest flexural strength (444 MPa) and elastic modulus (329 GPa). The effect of the carbon content on the hardness and toughness is not obvious. 

## 4. Conclusions

In this work, RBBC composites with 0–15 wt% free carbon content were prepared, and the influence of the carbon content on the microstructure and mechanical properties were studied. When the carbon content change is in the range of 0–10 wt%, the phase distribution in the sample is gradually uniform with the increase of the free carbon content. When the carbon content increases to 15 wt%, the phase distribution of the sintered sample begins to become uneven. In the C0 sample without free carbon, the B_4_C grains are obviously dissolved, the grains become facet-shaped, and the grain boundary becomes straight. In addition, many large-size grains were formed in the C0 samples. When the content of free carbon is 10 wt%, the phase distribution of RBBC-Si-infiltrated ceramic is uniform, and the residual silicon content is the lowest. Furthermore, the flexural strength (444 MPa) and the elastic modulus (329 GPa) of the C10 sample are the highest. The effect of the carbon content on the hardness and toughness is not obvious. The toughness of boron carbide ceramics obtained in this study is low, and the question of how to improve the toughness of boron carbide ceramics will be explored in the future.

## Figures and Tables

**Figure 1 materials-15-06028-f001:**
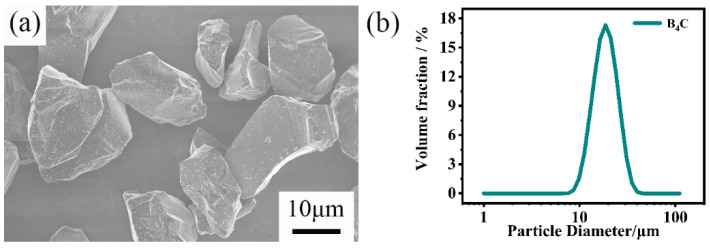
(**a**) SEM image and (**b**) particle size distribution of raw B_4_C powders.

**Figure 2 materials-15-06028-f002:**
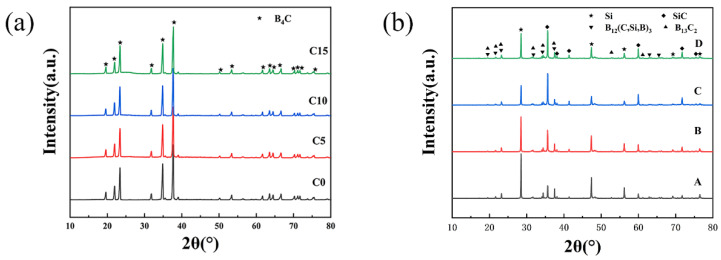
(**a**) XRD patterns of raw powder (mixture of B_4_C and C); (**b**)XRD patterns of composites with different contents of carbon.

**Figure 3 materials-15-06028-f003:**
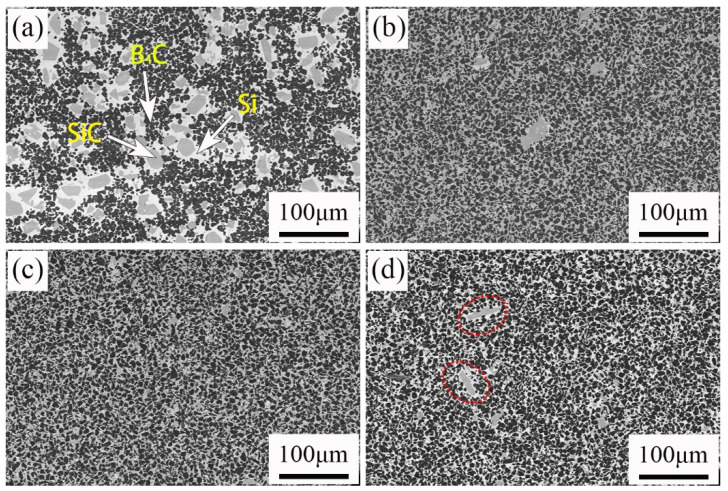
Backscattered SEM images of RBBC composites with different carbon contents. (**a**) 0; (**b**) 5 wt%; (**c**) 10 wt%; (**d**) 15 wt%.

**Figure 4 materials-15-06028-f004:**
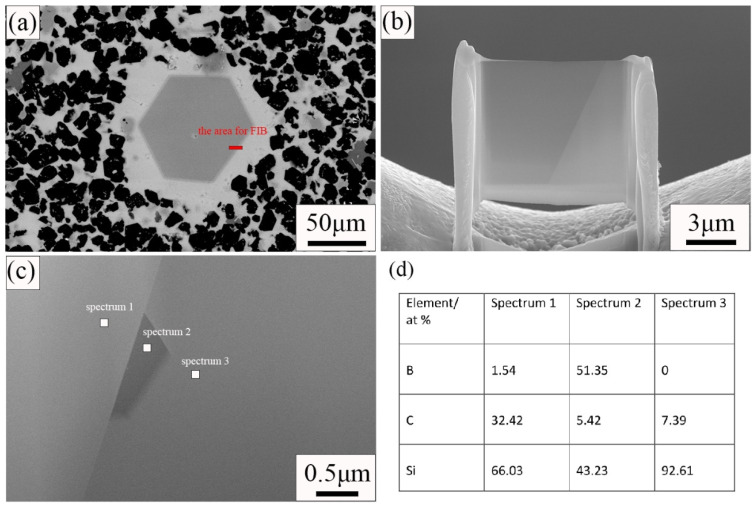
SEM and TEM images of C0 sample. (**a**) SEM image of C0 sample; (**b**) SEM image of sample cut by focused ion beam; (**c**) Dark field image of sample; (**d**) Results of EDS point scan.

**Figure 5 materials-15-06028-f005:**
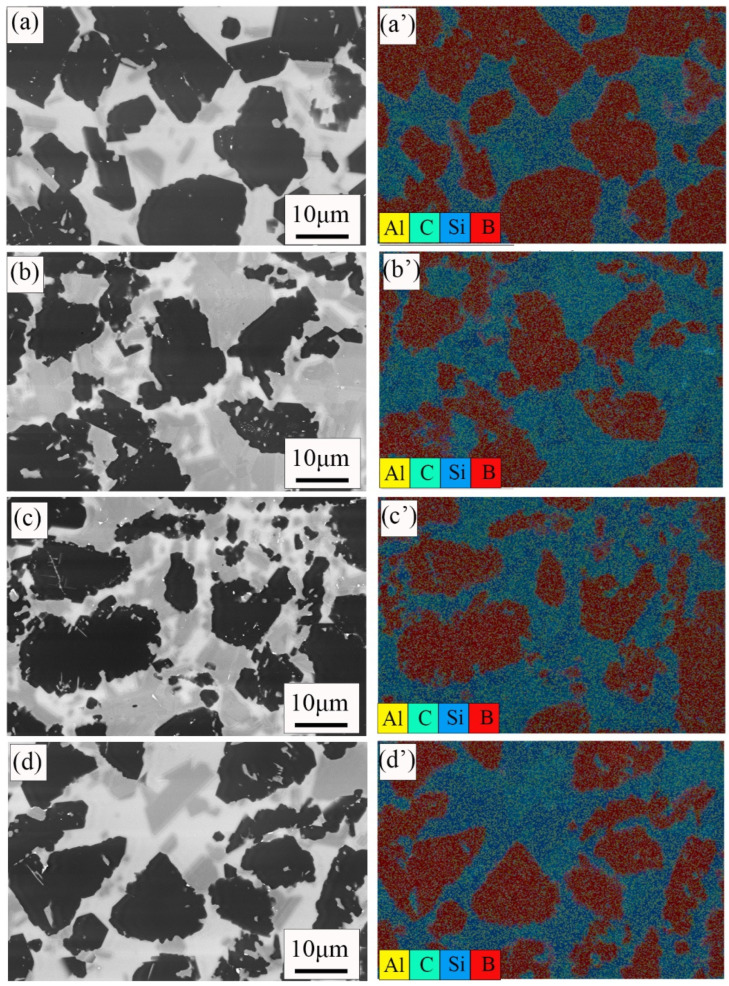
SEM images of the RBBC composites with different contents of carbon and EDS. (**a**,**a′**) 0; (**b**,**b′**) 5 wt%; (**c**,**c′**) 10 wt%; (**d**,**d′**) 15 wt%.

**Figure 6 materials-15-06028-f006:**
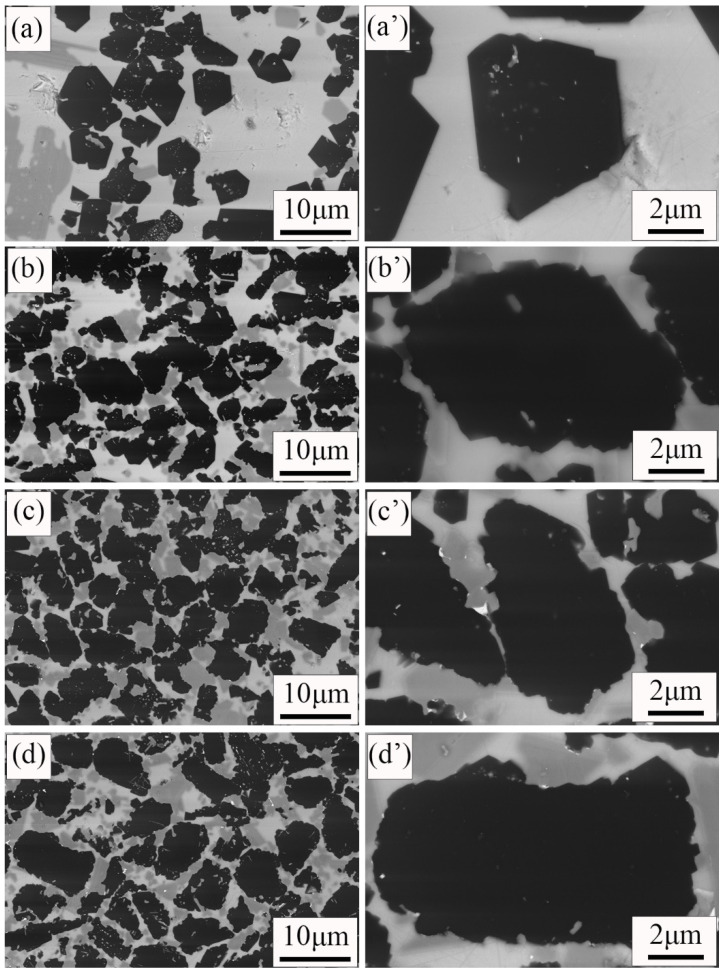
SEM images of the RBBC composites with different carbon contents. (**a**,**a′**) 0; (**b**,**b′**) 5 wt%; (**c**,**c′**) 10 wt%; (**d**,**d′**) 15 wt%.

**Figure 7 materials-15-06028-f007:**
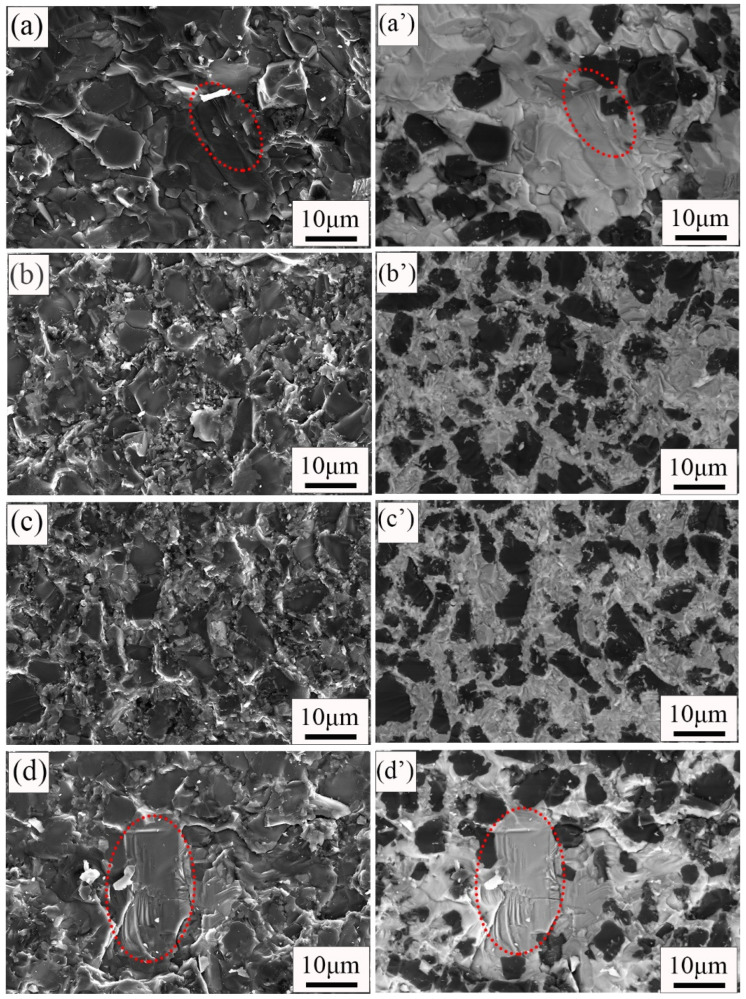
SEM images of RBBC composites with different carbon content additions. (**a**–**d**) are secondary electronic diagrams; (**a′**–**d′**) are backscattered electron diagrams.

**Figure 8 materials-15-06028-f008:**
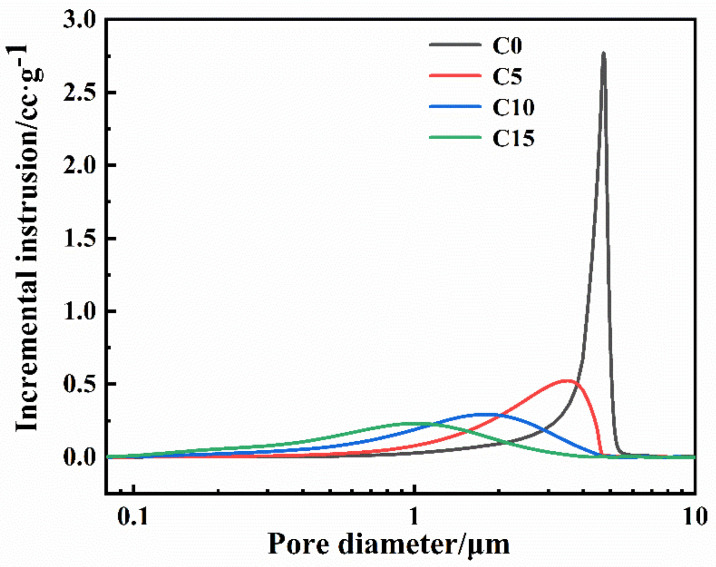
Pore size distribution of the green body with different carbon contents.

**Figure 9 materials-15-06028-f009:**
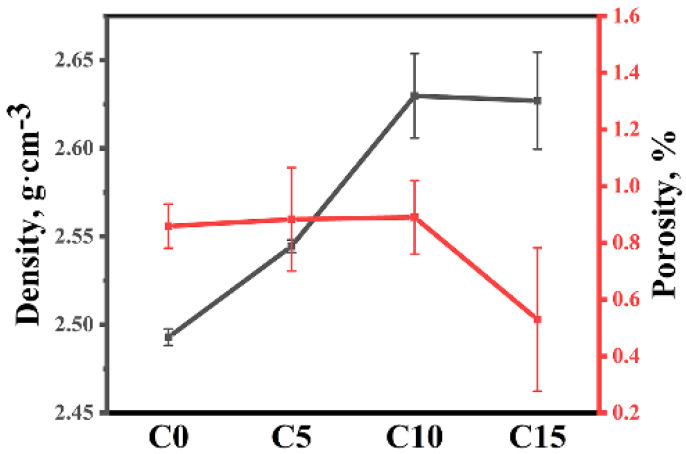
The open porosity and volume density of different RBBC composites.

**Figure 10 materials-15-06028-f010:**
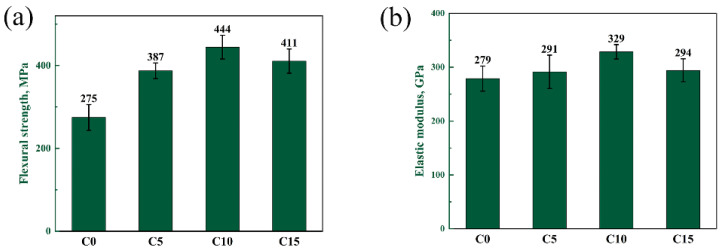
(**a**) Flexural strength and (**b**) elasticity modulus of composites with different carbon contents.

**Figure 11 materials-15-06028-f011:**
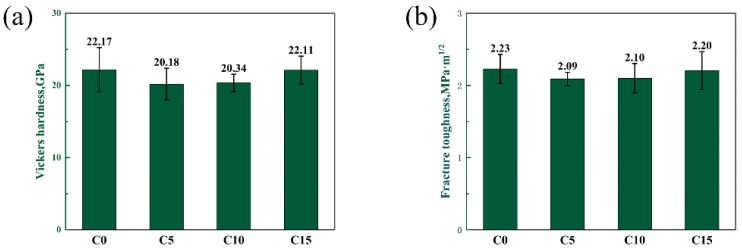
(**a**) Vickers hardness and (**b**) fracture toughness of composites with different carbon contents.

**Table 1 materials-15-06028-t001:** The nomenclature of composites and their raw material composition.

Samples	C0	C5	C10	C15
C/wt%	0	5	10	15
B_4_C/wt%	100	95	90	85

**Table 2 materials-15-06028-t002:** The residual silicon content of composites.

Samples	C0	C5	C10	C15
Si/wt%	27.18	19.57	14.14	16.70

**Table 3 materials-15-06028-t003:** The open porosity of green bodies.

Samples	C0	C5	C10	C15
Porosity/%	32.3	33.5	31.5	29.8

## Data Availability

Not applicable.

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
