# Peer review of "Effect of Carbon Content on Mechanical Properties of Boron Carbide Ceramics Composites Prepared by Reaction Sintering"

_materials, 2022, doi:10.3390/ma15176028_

Round 1

Reviewer 1 Report

Dear Sir,
The paper is interesting and well prepared however it can be improved in some areas as follows:

The title

It is confusing and should be rephrased.

In the abstract

Some grammatical error should be corrected and there are some typos in the manuscript for example:

-         It is recommended to begin this part with general sentence about the main topic to grasp the reader’s attention.

-         Please state exactly the performed analysis regarding the characterization such as XRD, SEM etc.

-         keywords should be arranged alphabetically and number of  keywords can  be increased to 5 terms.

In the other sections:

-         All chemicals and reagents being used in the study should be addressed in this section including where they are purchased or obtained.

-         State the model, the company and country of origin of any instrument used in the research such as XRD.

-         Could the author give more information about the XRD analysis.

-         Fig. 8 is not at a good quality and should be changed.

·        Future work should be mentioned at the end of the conclusion

Kindly, FORMAT the references correctly according to the author’s guide.

Regards

Reviewer 2 Report

The current manuscript studies the effect of carbon content on boron carbide – silicon composite microstructure and physical properties obtained using reaction sintering. I find the work interesting and detailed. However, it falls short in several areas especially in terms of procedural information that might be pertinent to the results. There are also some statements made in the paper which the data presented does not necessarily support. There are also grammatical errors throughout the manuscript which detract from understanding the document. I have divided my comments into major and minor comments below.

Major comments:

1.       Line 12 – 21: The abstract does not mention anywhere that the boron carbide ceramic composites are with ‘silicon’. Why not explicitly mention this? It is only much later in the paper that this becomes evident. Please mention this in both the abstract and introduction.

2.       Line 74: Figure 1 – There is no mention in the text of how these data were obtained. Please provide details of instruments and other relevant conditions in the text.

3.       Line 78: Was the Archimedes principle used only for sintered bodies? If so, please specify that. Also was a standard used for this measurement eg. ASTM? Please cite if used.

4.       Line 82: For three-point bend test, please provide details of the setup used and standards followed, if any.

5.       Line 83: Please provide equipment and standards information for fracture toughness measurements.

6.       Line 86: Please provide details of polishing, eg. grit size etc.

7.       Line 86-87: Please provide details of how the XRD peaks/phases were identified and what database/software was used.

8.       Line 103: It would be helpful to mark the two Si peaks in figure 2. Also, when the authors say that the intensity first increases and then decreases, is that with respect to the carbon content? Looking closely at figure 2b, the trends mentioned are not evident. If anything, the intensity of both the peaks first decreases and then increases with increasing carbon content from C0 to C15 which is the opposite of what the authors say. Please correct accordingly. Does this change affect the rest of the story? It would also be helpful if the authors provided a plot of the intensities of the two peaks as a function of the carbon content.

9.       Line 110: How was the silicon content for the different samples noted in table obtained? This needs to be described. Also, please include what the error on such a measurement would be.

10.   Line 112-113: Is the statement about generated silicon carbide and molten silicon filling into the pores of the green body common knowledge? Please provide appropriate reference. If not, please change language to include how the authors concluded this.

11.   Line 131: When the author’s say that the phases for the black, grey and white areas were confirmed by XRD, was XRD actually done on those specific areas? Or is the conclusion arrived at using the process elimination from data in figure 2 and 3?

12.   Line 152: Does figure 4 have enough information to conclude that the phase is B12(CxSiyBz)? I do not think there is enough information. Please provide more information or modify accordingly.

13.   Line 153: Please write down the reaction mentioned and provide appropriate references.

14.   Line 179-186: This entire discourse about shapes of grains could perhaps be quantified to help with the argument in terms of shape parameters such as sphericity etc.

15.   Line 218: Based on figure 8, seems like C15 might not satisfy this statement about majority grains being greater than 1 micron. Please modify accordingly. Would be helpful to have d50 values for pore size diameters for all the conditions.

16.   Figure 9: Interesting that the variabilty in the density measurement increases drastically at C10 even though the porosity does not change that much. What might the reason for this be?

17.   Line 266-267: Figure 11 and author's assertion in line 254-259 does not support this assertion about the sample with 0% free carbon having highest hardness and toughness. They all seem to be same. Please remove this statement.

18.   Line 280-282: These two statements about hardness and toughness trends are contradictory. One condition cannot be highest if there is no significant trend. Please modify accordingly.

Minor comments:

1.       Line 3: Suggested to say ‘boron carbide ceramic composites’ instead of ‘boron carbide composites’ to make the title more representative.

2.       Line 25: Use ‘boron carbide ceramics are a significant’ instead of ‘is a’. Similar corrections from singular to plural in the rest of the introduction, eg. ‘It has’ in line 25, ‘it is widely’ in line 27.

3.       Line 29: Missing comma after ‘material cutting’.

4.       Line 43-44: Please provide references for the recent works on reaction sintering of boron carbide ceramic composites.

5.       Line 49: ‘but did not focus on consider’ makes no sense grammatically. Please rephrase.

6.       Line 57: Use ‘commercially’ instead of ‘commercial’.

7.       Line 58: Please add the source for carbon black.

8.       Line 66: ‘respectively’ is redundant. Please remove.

9.       Line 72: Missing space after ‘vacuum’.

10.   Line 79: Please use ‘green body’ instead of ‘green’.

11.   Line 79: Need more details on the ‘poremaster60’. Is this nitrogen porosimetry? Mercury? Please also provide manufacturer information.

12.   Line 83: Remove capitalization for ‘Fracture’.

13.   Line 124: What is the error on the porosity measurement? Please include it.

14.   Line 126: Figure 3 – The red text inside the figures is not legible.

15.   Line 137: Rephrase to ‘when….then’.

16.   Line 142: Why not specify SiC when mentioning ‘abnormally large grey grains’? Please modify accordingly.

17.   Line 143: Missing space after end of sentence.

18.   Line 145: Rephrase to ‘using a focused ion beam’. Make sure to not capitalize.

19.   Line 158: Use ‘are formed by’ instead of ‘is formed by’.

20.   Line 161: Use ‘C is provided’ instead of ‘C provided’.

21.   Figure 5: The legend text in the EDS images is not legible. Please improve.

22.   Line 177: Change ‘bule’ to ‘blue’.

23.   Line 180: Change ‘curve’ to ‘curved’.

24.   Line 188: Use ‘Mechanical properties’ instead of ‘mechanical’.

25.   Line 215-216: Awkward construction. It is difficult to understand what the authors are saying. Please rephrase.

26.   Line 228: Change ‘frist’ to ‘first’.

Reviewer 3 Report

The authors submitted a well-written article entitled 'Influence of Carbon Content on Microstructural and Mechanical Properties of Boron Carbide Ceramics Synthesized by Reaction Sintering'. The general concept of performed studies is well-designed, what only increases the scientific value of the paper. Only a few typos and minor editorial mistakes have been detected in text, like:

lines 85-86, 195-196, 200-201, 241 - 245 - different font size

Moreover, authors don't add what kind of TEM they use.

Considering mentioned above, I suggest publishing a paper after improving these minor mistakes.

Author Response

Response to Reviewer 3 Comments

Thank you very much for your comments on our manuscript entitled “Influence of Carbon Content on Microstructural and Mechanical Properties of Boron Carbide Ceramics Synthesized by Reaction Sintering”. According to your suggestion, we have revised the article. We have corrected the font errors in the article and added information about TEM in the Characterization section.